# Bullying and Health Related Quality of Life among Adolescents—A Systematic Review

**DOI:** 10.3390/children9060766

**Published:** 2022-05-24

**Authors:** Viney Prakash Dubey, Justina Kievišienė, Alona Rauckiene-Michealsson, Sigute Norkiene, Artūras Razbadauskas, Cesar Agostinis-Sobrinho

**Affiliations:** Faculty of Health Sciences, Klaipeda University, 92294 Klaipėda, Lithuania; justina.kievisiene@gmail.com (J.K.); alona.rauckiene-michaelsson@ku.lt (A.R.-M.); sigute.norkiene@gmail.com (S.N.); arturas.razbadauskas@gmail.com (A.R.); cesaragostinis@hotmail.com (C.A.-S.)

**Keywords:** bullying, health related quality of life, adolescents

## Abstract

Health-related quality of life is among global health goals not only in adulthood but also in childhood and adolescence. Being a multi-component construct, health-related quality of life covers various domains, such as physical and psychological wellbeing and social and environmental areas. Bullying might significantly influence those domains especially in adolescence, a period of life when numerous personal and interpersonal transformations are experienced. Therefore, the aim of the current systematic review was to provide a comprehensive overview of the relationship of bullying with the health-related quality of adolescents’ lives. An electronic literature search was performed using PubMed, Embase, and Cochrane Library, and 3621 full-text articles were identified. After a selection process, 12 studies covering diagnosis, prevention and treatment for each of the three sections “adolescents”, “health related quality of life” and “bullying” were reviewed. An overall reduction in health-related quality of life in regard to bullying appeared from the studies analyzed, as well as a decline in adolescent mental health. Different bullying types were identified as causing harm to various adolescents‘ health-related quality of life domains. These findings may contribute to effective bullying management in schools and/or societal settings, and inform intervention strategies for maintaining the quality of life of adolescents being bullied.

## 1. Introduction

Health-related quality of life (HRQoL) is a multidimensional construct that contains various domains such as physical, psychological, social, and environmental [1,2]. The World Health Organization (WHO) places significant emphasis on wellbeing and HRQoL as global health objectives, particularly among adolescents, and emphasizes the need for research to identify the primary causes of health problems in this age group [3]. Additionally, the WHO states that adolescents’ health and wellbeing are critical for the health and sustainability of societies [4]. 

HRQoL in adolescents has often been researched alongside that in children or adults as there is a change around the onset and beyond puberty [5] marked by numerous transformations in the body, mind, and behavior [6]. Adolescence is characterized by biological, social, and behavioral changes that affect lifestyle, social, family, cultural, and spiritual interactions, as well as a sense of self-identity [1]. It is also the period of exposing oneself to risky scenarios that might have serious impacts in relation to HRQoL [7,8]. During adolescence, peers often have a larger influence on each other than they had in childhood, or would have later in life when the individual reaches adulthood [9]. Throughout this period of life, peers play an essential role in either promoting or hindering healthy psychological and social development. Impacts may be both positive and negative; on the positive side, peers provide examples and comments that adults do not provide to adolescents [9,10]. On the other hand, if a teenager is despised or, worse, rejected or tormented by peers, adolescence may become one of the loneliest times in life and can negatively influence the HRQoL [8].

Academic research has stressed bullying as a form of violence, defined as harshness, intimidation, and/or aggressiveness [11], and it may be categorized into types. The first is “direct physical”, marked by actions that include attacking physically, stealing or destroying other people’s property [12]. “Direct verbal” may be described as any form of verbal, racial, or sexual harassment [13], and “indirect” bullying behaviors cover systematically excluding a person via gossip or spreading rumors, threatening to exclude someone from a group to gain favor, or manipulating the social life of another person in general [14]. School bullying victims are more prone to experience psychological discomfort, depressive symptoms, self-harm, and even suicide [15]. As its influence on the development of children and adolescents can have catastrophic physical, psychological, and social implications, the high prevalence of school violence at all levels is a global concern and a major public health issue [16].

Bullying is a major social concern that extends beyond the academic and personal levels, severely influencing adolescent HRQoL and, in most cases, causing irreversible harm [17]. A systematic review has explored bullying prevalence or its health impacts, but not HRQoL [18]. Several cross-sectional studies have showed a link between bullying and HRQoL [1,7,11,17,19]; however, there are no systematic reviews that focus on bullying and its relationship with HRQoL among adolescents, to our knowledge. Taking this into account, it was important to conduct a systematic review of the literature that highlights the association of bullying with the health-related quality of adolescents’ lives. Therefore, the aim of this systematic review was to provide a comprehensive overview of the relationship between bullying and the health-related quality of adolescents’ lives.

## 2. Methods

### 2.1. Eligibility Criteria

Inclusion criteria: School-going 8–18 year old children and adolescents; longitudinal and cross-sectional studies.

Exclusion criteria: Children younger than 8 years old; case studies, review articles, dissertations, letters, editorials, book chapters, qualitative studies, and conference abstracts, as well as articles written in a language other than English.

### 2.2. Information Sources

We searched the following databases from the beginning to 21 December 2021, only for articles published in English language:

► Cochrane Central Register of Controlled Trials, ► MEDLINE (OvidSP, 2011 to 21 December 2021), ► Cumulative Index to Nursing and Allied Health Literature (CINAHL) (EBSCO, 2011 to 21 December 2021), ►Embase (OvidSP, 1980 to 21 December 2021), ► PsycINFO (OvidSP, 2011 to 21 December 2021), ► Cochrane Complementary Medicine Field Trials Specialized Register (Cochrane Register of Studies Online (CRSO)), ► Allied and Complementary Medicine Database (AMED) (OvidSP, 2011 to 21 December 2021), ► CBN Trials Register (Cochrane Register of Studies (CRS)), ► IndMED, ►Web of Science, ► US National Institutes of Health ClinicalTrials.gov, ► PubMed.

This systematic review protocol has been registered with PROSPERO (CRD42022326768) and is available upon request.

### 2.3. Search Strategy

We searched for specific words in the title and abstract, as well as Medical Subject Headings (MeSH) terms. We combined words of interest (e.g., ‘Adolescent (MeSH)’ OR ‘bullying’ OR ‘adolescent’) AND type of school violence (e.g., ‘fighting (MeSH)’ OR ‘gang violence and bullying (MeSH)’ OR ‘gang-violence’ OR ‘school-bullying’ OR ‘teenage-bullying’ OR ‘bullying-teenage’) AND association of bullying and quality of life (e.g., ‘quality of life (MeSH)’ OR ‘health related quality of life’ AND ‘HRQoL’ OR ‘bullying and health related quality of life’, respectively) OR prevalence domains (e.g., ‘Prevalence-bullying (MeSHs)’ OR ‘Increase bullying’).

### 2.4. Selection Process

A three-stage process was used to select the studies to be included, with two reviewers independently evaluating each identified citation to determine whether or not it should be included. The initial phase consisted of the evaluation of titles selected through the above-described systematic searches. If the title included the words “adolescent” and/or “bullying”, the article was listed in the first screen. The abstracts of all articles that met the search criteria were then reviewed. Both reviewers independently retrieved and read full-text articles that met the inclusion criteria and assessed their suitability for inclusion in the study. Disagreements between the reviewers were resolved by consensus or by the decision of a third, independent reviewer.

### 2.5. Data Extraction and Critical Appraisal

Extracted data were systematized using a specifically designed standardized data extracting form (see study protocol), and afterwards, the reviewers compared the extracted data for consistency. For critical appraisal and to limit bias, each included study was independently examined, and a modified version of the Newcastle Ottawa cohort scale was employed for cross-sectional research (Appendix A) [20]. To resolve differences, consensus was used; results are presented in Table 1. General study information, characteristics of participants and interventions, withdrawals, and outcome measures were extracted. Table 2 presents a full list of extracted data items. Where data were not available from tables or the results section, the authors of the relevant study were contacted via email, with a follow-up email sent two weeks later if they did not respond to the initial email.

## 3. Results

### 3.1. Study Selection

We identified 3621 articles by searching the databases MEDLINE (via PubMed), Embase (Ovid), and Web of Science. From that total, 1204 studies were excluded due to duplication, while 876 were classified as ineligible by automation tools. Another 761 studies were removed for additional reasons. We examined the titles and abstracts of 1541 studies and excluded 1256 based on our findings. We screened 165 articles for eligibility. Twelve studies were included in the review. A PRISMA-compliant flowchart shows the process for selection of relevant studies from the databases (Figure 1).

### 3.2. Study Characteristics

The methodological details of the included studies are given in Table 2. The characteristics of study populations, intervention protocols and outcome measures and tools of the measures are briefly described. All 12 selected studies, all cross-sectional in their design and published between 2016 and 2021, were included in this systematic review.

## 4. Discussion

The present systematic review summarizes the association between HRQoL and bullying among children and adolescents. The findings showed that bullying can significantly influence HRQoL among members of this population, mostly measured by well-adopted and worldwide-recognized KIDSCREEN [27] questionnaires. Overall lower health-related quality of life scores were identified in all studies, with such most affected domains as physical and psychological wellbeing, social relations (parents/peers) and school environment. Furthermore, differences in HRQoL were observed with regard to the different types and prevalence of bullying among genders, as well as their impact on HRQoL. Recognizing specificities and tendencies of bullying, and identifying the most vulnerable adolescents affected by it, provides guidelines for effective prevention strategies in school, societal levels and public health.

The current systematic review showed similar bullying tendencies in different countries such as Sweden [21], England [8], Spain [17,23,24,26], Mexico [22], Brazil [7], Norway [2], China [16], and Vietnam [25]. Additionally, some studies highlighted that girls more often reported being bullied than boys, but that the differences were not statistically significant [2,11,21], whereas various studies indicated that boys are bullied more than girls [1,13,28,29]. However, the cross-sectional nature of these studies did not make it possible to establish causal relations, and tentative conclusions were only drawn on a theoretical basis. Moreover, different questionnaires were used between studies, which created a lack of substantial supporting evidence to confirm. In this systematic review, we also found that the playground (patio) was the most frequently bullied location, followed by the classroom and school [7,19]. It might be the case that in both, the playground and the classroom, where there are staff and teachers, bullying occurs in disguised form in a variety of situations, which is difficult to address. Non-physical violent behaviors, such as verbal abuse or social abuse, are more difficult to recognize and avoid [25]. Adolescence is a time when youngsters are exposed to a range of risk situations that might have serious consequences later in life, and the school is an essential social institution for education and life preparation. A broader educational approach that promotes students’ better social and emotional development should be implemented attentively [1] in order to prevent bullying and its negative impact on adolescent life. Addressing a number of factors such as the student–teacher interaction, school policies, and communities [28] might also contribute to improving the quality of life of kids experiencing harassment.

Among bullying types, age seems to be an important factor. A considerable shift in violent behaviors with age was shown by previous studies [21,25]. In this review, we found that physical violence was more prevalent in elementary schools; but, at higher education levels, such as in middle and high schools, teens are more likely to experience bullying as a consequence of being placed under pressure by others, being jealous of others, or being gossiped about by others [25]. This behavioral change can be explained by the view that as people grow older and more educated, then the most typical type of aggression moves from direct to indirect expression [30].

Different sorts of bullying might have different effect on adolescents’ HRQol. Earlier studies revealed that physical bullying is the most destructive [13,15], while social victimization has been found to have the greatest impact on adolescents’ psychological wellbeing [30] and HRQoL [1]. According to the findings of the current systematic review, the most common sort of bullying, which worsened the HRQoL of adolescents in England [8], Brazil [7], Mexico [22] and Sweden [21], was the verbal form of aggression, which included insulting and name-calling. Among Vietnamese adolescents [25], the most common forms of bullying were social aggression, followed by verbal bullying, physical bullying, and sexual bullying. Among Chinese adolescents [16], traditional bullying (which includes direct aggression such as physical and verbal or indirect aggression such as spreading rumors, isolation, theft of belongings, and destruction [18]) was the most common, and among Spanish [23,24,26] and Norwegian [2] adolescents, physical and verbal/social bullying were the most common forms associated with worse HRQoL scores. Bullying involves insults that outweigh physical aggressiveness such as kick-punching, punching, and the threat of injuring a peer in school [9]. Therefore, it is important to recognize the threat of indirect bullying behaviors (e.g., social exclusion, social aggression, etc.), that are typically seen as less serious than physical bullying behaviors [14].

Bullying of any kind has negative and long-lasting impacts on young people’s health and wellbeing [1]. Our study results summarize that despite the type, bullying significantly decreases the HRQoL among adolescents [2,11,31], and that adolescents who were bullied repeatedly reported that it had a greater impact on their daily activities and social lives than did those who were not actively involved in bullying [21,32]. Furthermore, a negative impact of bullying on individuals’ moods and emotions [30,33] may also contribute to declines in HRQOL [22]. Due to bullying, adolescents feel shame, neglected by their social network/peers, and guilt, which lowers their self-esteem and self-perceptions [14]. Similar findings were identified in this systematic review. For example, a study on Mexican adolescents [22] showed that bullied victims felt neglected by their peers and had low self-perceptions. Another disclosed bullying and depressive moods [33]. A few studies revealed an increased tendency among victims to attempt suicide [17,21], and one highlighted the high level of anxiety after being bullied [25].

It is unclear what causes some adolescents to be bullied repeatedly. However, studies have show that individuals who exhibit behavior that differs from that of their peers are more likely to be bullied [12,14,29,33]. Additionally, adolescents who reported being bullied often lack social skills, may have a weak family environment [32] or negative peer networks, or have insufficient personal resources, which leaves them more vulnerable to being bullied [15]. Teens who are bullied may be more susceptible to bullying because they are more sensitive to peer behavior or suffer from mental health issues [14]. Identifying adolescents who exhibit these qualities may prove to be an effective strategy for dealing with the problem. The main limitation of the present study may be regarded as social desirability bias, which may have resulted from adolescents’ tendency to answer favorably. However, the use of self-reporting surveys is widespread, necessary, and efficient for gathering data from a large number of individuals. The determination of who is a victim of bullying is dependent on the instrument employed, recollection period, and the concept of bullying, despite decades of research into the topic.

## 5. Conclusions

To conclude, this review presented empirical evidence of a significant decline in HRQoL among adolescents who are being bullied. In order to effectively prevent bullying, the needs of adolescents who have been bullied must be taken into consideration, and bullying victims must be given top priority in order to ease the harm caused to their health-related quality of life. Different bullying types were identified as causing harm to various adolescents health-related quality of life domains. These findings may contribute to effective bullying management in schools and/or societal settings, and inform intervention strategies in maintaining the quality of life of adolescents being bullied.

## 6. Limitations of the Study

Despite the fact that the findings of this systematic review expanded our understanding of the associations between bullying and health-related quality of life, it is important to note that it has some limitations. The cross-sectional nature of the reviewed studies did not allow us to establish causal relations, and tentative conclusions were only drawn on a theoretical basis. Furthermore, there would have been inconsistencies in the analysis as a result of methodological differences in the way bullying victimization was defined and measured across the studies. This is because there is no general agreement on the most accurate method to measure bullying victimization, so there is no single method that is universally accepted. This systematic review revealed different type of instrumental tools to measure the prevalence of bullying among adolescents. However, we lacked data to measure and represent bullying elements such as intentionality, repetitiveness, and differences in power.

## Figures and Tables

**Figure 1 children-09-00766-f001:**
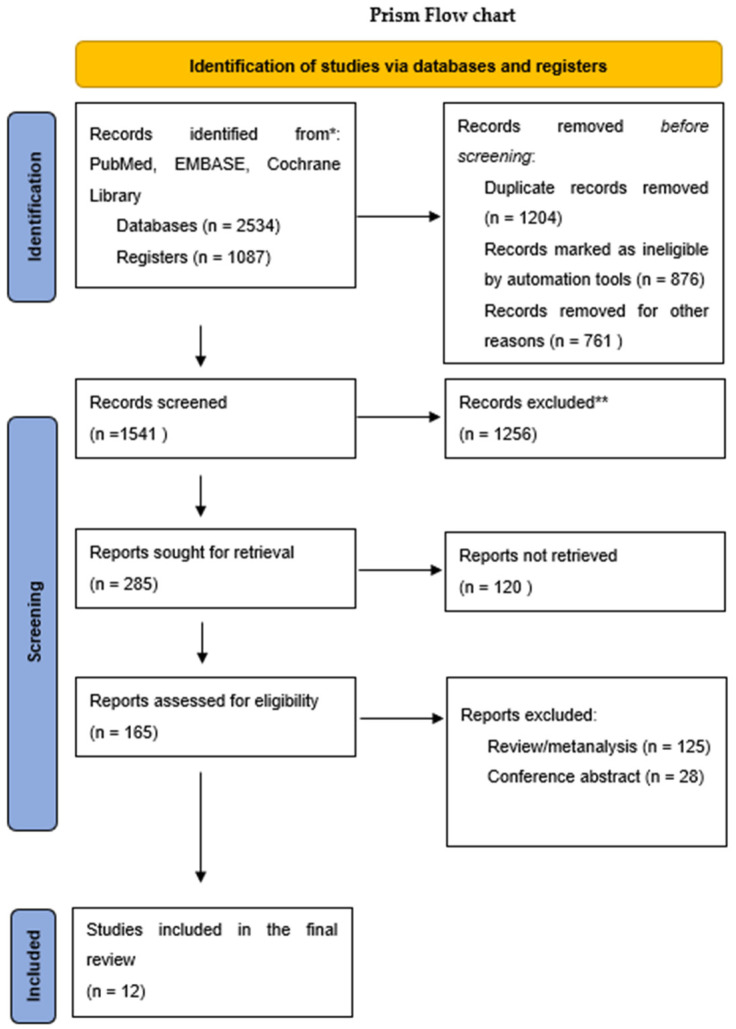
Search and selection process. (* reported the number of records identified from each database or register searched (rather than the total number across all databases/registers). ** records were excluded by automation tools.)

**Table 1 children-09-00766-t001:** Results of the critical appraisal of the included studies. (+ for self-reported outcomes; ++ for the outcome with independent blind observers).

Author (Year)	Study Design	Selection				Comparability	Outcome	
		Representative of the Sample	Sample Size	Non-Respondents	Ascertainment of Exposure	Based on Design and Analysis	Assessment of Outcome	Statistical Test
Beckman et al. (2016) [21]	Cross-sectional	+	+	+	+	+	+	+
Chester et al. (2017) [8]	Cross-sectional	+			+	+	+	+
Diaz et al. (2017) [17]	Cross-sectional	+	+		+		+	+
Fantaguzzi et al. (2017) [19]	Cross-sectional		+		++	+	+	+
Hidalgo-Rasmussen et al. (2018) [22]	Cross-sectional	+			+		+	+
Albaladejo-Blázquez et al. (2019) [23]	Cross-sectional	+	+	+	+	+	+	+
Garbin et al. (2019) [7]	Cross-sectional	+		+	++	+	+	+
Haraldstad et al. (2019) [2]	Cross-sectional	+	+		+	+	+	+
González-Cabrera et al. (2020) [24]	Cross-sectional	+	+	+	+		+	+
Xiao et al. (2021) [16]	Cross-sectional	+			+	+	+	+
Ngo et al. (2021) [25]	Cross-sectional	+			+		+	+
Martin-Perez et al. (2021) [26]	Cross-sectional	+	+		++	+	+	+

**Table 2 children-09-00766-t002:** Reviewed studies of HRQOL and bullying according to sample, method, results, and conclusion.

Author (Year)	Country	Sample Size (% Female)	Age Category	Research Instruments	Statistical Model	Findings
Beckman et al. (2016) [21]	Sweden	758 (56.6%)	15–17	Self-reported; Short-Form Health Survey SF36 and SF 6D, (set of questions on bullying)	Linear regression analyses	Mean preference-based health-related quality of life scores were 0.76 for non-victims and 0.70 (0.69–0.71) (95 % CI); *p* < 0.01 for victims of bullying, respectively.
Chester et al. (2017) [8]	(HBSC) study, England	5335 (48.5%)	11–15	Self reported; Revised Olweus Bully/Victim Questionnaire; HRQoL. KIDSCREEN-10	Linear regression analyses	Weekly relational bullying resulted in an estimated 5.352 (95% confidence interval (CI), −4.178, −6.526) decrease in KIDSCREEN- 10 score compared with those not experiencing relational bullying. Bullying is associated with lower HRQoL. Girls were more likely than boys to report relational bullying, but the negative association with HRQoL was the same for both genders.
Diaz et al. (2017) [17]	Spain	769 (46%)	13–17	Self-reported social media use (set of questions), HQRoL: Kidscreen-52, 22-item Victimization Peers Scale	Pearson’s correlations; linear regression	Bullying at school has a negative effect on a child’s HRQoL, and the effect is bigger for aggressive victims than for pure victims.
Fantaguzzi et al. (2017) [19]	England	6667 (51.8%)	11–12	Quality of life was measured using both the Child Health Utility 9D, CHU-9D and Pediatric Quality of Life Inventory, PedsQL instruments, The Gatehouse Bullying Scale (GBS).	Linear regression analyses	When compared to their classmates, children who were bullied or acted aggressively at school had lower health-related quality of life and utility scores. The difference was −0.1 on a scale of 0–1 for CHU-9D utility scores and −16 on a scale of 0–100 for PedsQL scores.
Hidalgo-Rasmussen et al. (2018) [22]	Mexico	248 (52.9%)	8–18	Self-reported social media use (set of questions), HRQoL using KIDSCREEN 10, Bullying using social acceptance domain of KIDSCREEN-52 questionnaire	Multivariate logistic regression	After adjusting for health perception, gender, and age, being a victim of bullying doubled the risk of having a lower HRQoL than not being a victim, OR 2.3%. (1.7–3.1).
Albaladejo-Blázquez et al. (2019) [23]	Spain	1723 (49%)	11–19	Self-reported social media use (set of questions), Bullying: The Illinois Bully Scale, Health-Related Quality of Life (HRQoL): KIDSCREEN-27, The Homophobic Verbal Content Bullying: Homophobic Content Agent Target (HCAT) Scale, Depression: Patient Health Questionnaire (PHQ-9), Anxiety: Generalized Anxiety Disorder-7 (GAD-7)	ANCOVAs	Both traditional and homophobic bullying negatively affect adolescent health-related quality of life and mental health. Physical wellbeing [Role: F (3,1723) = 4.751, *p* = 0.003, η2 = 0.008], psychological wellbeing [Role: F (3,1723) = 18.808, *p* = 0.0001, η2 = 0.032], autonomy and parent relations [Role: F (3,1723) = 14.574, *p* = 0.0001, η2 = 0.025], social support and peers [Role: F (3,1723) = 10.128, *p* = 0.0001, η2 = 0.017], school environment [Role: F (3,1723) = 25.358, *p* = 0.0001, η2 = 0.042].
Garbin et al. (2019) [7]	Brazil	382 (62%)	11–16	Self-reported social media use (set of questions), The Victimization and Peer Aggression Scale (VPAS) for bullying, quality of life of adolescents, the WHOQOL-Bref instrument	Spearman correlation	Study concluded that as bullying increases, the health-related quality of life decreases. Physical (−0.26; *p* = 0.001) Psychological (−0.38; *p* = 0.001) Social Relations (1.9; *p* = 0.001) Environment (−0.16; *p* = 0.001)
Haraldstad et al. (2019) [2]	Norway	723 (54%)	12–18	Self-reported social media use (set of questions), KIDSCREEN-52 questionnaire, Bullying questions: “How often have you been bullied?” and “How often have you bullied others?” Both questions were answered using a five-point scale where 1 = “never”, 2 = “only once or twice”, 3 = “two or three times a month”, 4 = “about once a week”, and 5 = “several times a week”. In the analyses presented here, the two bullying variables were dichotomized as never or only once or twice or more.	Pearson’s correlation	Being involved in bullying as a victim is correlated with poorer health-related quality of life. Phys wellbeing (−0.07; β = 0.05); Psych wellbeing (−0.19; β = −2.24); Mood/emotion (−0.20; β = −2.03); Self-perception (−0.14; β = −1.38); Autonomy (−13; β = −1.71); Parent relationship (−14; β = −1.22); Financial resources (−22; β = −3.53); Peers and social supp (−0.23; β = −3.34); School environment (−0.14; β = −3.82).
González-Cabrera et al. (2020) [24]	Spain	12,285 (50.3%)	11–18	For the evaluation of HRQoL, the Spanish version of the KIDSCREEN-27, Spanish version of the European Bullying, Cyberbullying Triangulation Questionnaire-CTQ Intervention Project Questionnaire (EBIPQ)	Pearson correlations; Anova	Study concluded that bullying resulted in lower health-related quality of life; with respect to age differences, study showed that involvement in bullying problems increases with age. Correlations showed between the Bullying and the five KIDSCREEN-27. Physical wellbeing (−0.143), Psychological wellbeing (−0.326), Autonomy and parent relation (−0.245), Social support and peers (−0.231), School environment −0.277 Among boys and girls: t (*p*)− 1.03 (0.304) d = 0.02
Xiao et al. (2021) [16]	China	2155 (50.1%)	9–17	Olweus Bully/Victim Questionnaire, HRQoL The KIDSCREEN 10 tool was used to assess self-reported HRQoL.	Multivariate logistic regressions	Being a victim of traditional bullying increases the risk of having lower health-related quality of life. (β = −3.55, *p* < 0.001, SE = 0.41) Students’ health-related quality of life scores decreased with the increase in the degree of traditional bullying. (β = −10.28, *p* < 0.001, SE = 1.19)
Ngo et al. (2021) [25]	Vietnam	712 (58.1%)	11–14	Bullying; self-reported using questions, HRQoL of students was assessed by utilizing the Vietnamese version of EuroQol-5 dimensions-5 levels (EQ-5D-5L), The Depression, Anxiety, and Stress Scale—21 Items (DASS-21)	Logistic regression model; multivariate Tobit regression.	Being bullied was significantly correlated with the decrement of HRQoL, and a higher risk of depression, anxiety, and stress among adolescents. Physical bullying (Coef. = −0.05; 95%C = −0.08–−0.02), social aggression (Coef. = −0.02; 95%C = −0.04−0.01) and verbal bullying (Coef. = −0.03; 95%C = −0.06–−0.01)
Martin-Perez et al. (2021) [26]	Spain	1411 (47.2%)	12–18	HRQoL was assessed by Spanish KIDSCREEN-52 questionnaire. Bullying was reported using Spanish version of the validated Adolescent Peer Relations Instrument-Bullying (APRI)	Multivariate logistic regression	Physical, verbal/social bullying were closely associated with worse scores in all the HRQoL subscales and *Social Acceptance* was the most affected dimension, followed by *Moods and Emotions*, which, in the most serious cases of harassment, had a 15.86 and 8.35 (*p* < 0.001) higher risk.

## Data Availability

Data used in this study are available in Table 1.

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
