# Peer review of "Bullying and Health Related Quality of Life among Adolescents—A Systematic Review"

_children, 2022, doi:10.3390/children9060766_

Round 1

Reviewer 1 Report

Dear authors

I am pleased with your idea to contribute to the field with this kind of work. Even though it seems you have done an important effort, a lot of work and reflection are needed in order to make this work suitable for scientific publication. 

I first want to remind you that systematics reviews seek to apply scientific strategies that limit bias to the systematic assembly, critical appraisal, and synthesis of all relevant studies on a specific topic (see Greener & Greenshaw, 1996). And, from my perspective, your work fails to accomplish this goal. 

Scholars in the field certainly want to know the overall effects of bullying on individuals' health, however, the arguments you are using lack an in-depth reflection. Let me give you an example. The scholars assert girls reported--across studies conducted around the globe--being more bullied than boys; however, they never discuss the validity of such comparisons. That is, in specific, the scholars did not explain whether those results come from studies using invariant scales (by gender, age, stage of adolescence, etc.). Therefore, such statements are at best questionable, especially when even UNESCO has acknowledged the difficulty to make such comparisons.

In that regard, the variety of instruments used by the studies reported is not equally measure bullying and its three specific elements: intentionality, being repetitive, and differences of power. 

Furthermore, you may guide your study following the steps proposed by Magarey (2000) since there are important pieces missing in your manuscript (searching and appraising current literature (from XX to XX; data extraction; analysis and synthesis).

Overall, this manuscript has to be matured in order to be suitable for publication. 

You may see the UNESCO report at https://unesdoc.unesco.org/ark:/48223/pf0000366483

Author Response

Dear authors

I am pleased with your idea to contribute to the field with this kind of work. Even though it seems you have done an important effort, a lot of work and reflection are needed in order to make this work suitable for scientific publication. 

Authors: Thank you for the positive feedback regarding our study, as well as the constructive input.  We have addressed the concerns you have raised. 

I first want to remind you that systematics reviews seek to apply scientific strategies that limit bias to the systematic assembly, critical appraisal, and synthesis of all relevant studies on a specific topic (see Greener & Greenshaw, 1996). And, from my perspective, your work fails to accomplish this goal. 

Authors: We thank the reviewer for this suggestion.  We did an update .We apologize for this oversight. We have included the risk of bias assessment chart which is highlighted in the manuscript using modified newcastle - ottawa quality assessment scale.

Scholars in the field certainly want to know the overall effects of bullying on individuals' health, however, the arguments you are using lack an in-depth reflection. Let me give you an example. The scholars assert girls reported--across studies conducted around the globe--being more bullied than boys; however, they never discuss the validity of such comparisons. That is, in specific, the scholars did not explain whether those results come from studies using invariant scales (by gender, age, stage of adolescence, etc.). Therefore, such statements are at best questionable, especially when even UNESCO has acknowledged the difficulty to make such comparisons.

Authors:  Thank you for raising this question, we have added some text to address this review concern. Although all the reflections are given based on the studies included in this study,

In that regard, the variety of instruments used by the studies reported is not equally measure bullying and its three specific elements: intentionality, being repetitive, and differences of power. 

Authors:  Thank you for point this out. Regarding instruments, we add some text in the limitation session as well to the discussion to address this point

Furthermore, you may guide your study following the steps proposed by Magarey (2000) since there are important pieces missing in your manuscript (searching and appraising current literature (from XX to XX; data extraction; analysis and synthesis).

Overall, this manuscript has to be matured in order to be suitable for publication. You may see the UNESCO report at https://unesdoc.unesco.org/ark:/48223/pf0000366483

Authors:  Thank you for your suggestion, we have added necessary arguments in the manuscript

Reviewer 2 Report

Appreciate the authors' presenting a well-written paper. The authors may want to explore limitations of this SR. Consider, at the minimum, talking about gender differences to address our social climate of equity.

Few suggestions in the paper I have attached.

Thank you for addressing an important topic.

Author Response

Appreciate the authors' presenting a well-written paper. The authors may want to explore limitations of this SR. Consider, at the minimum, talking about gender differences to address our social climate of equity.

Authors: Thank you for your time for reviewing our manuscript. Thank you for positive appraisal our work. We have addressed the suggestions. 

Few suggestions in the paper I have attached.

Authors: We have gone through the suggestions and included them in manuscript.

Thank you for addressing an important topic.

Authors: Thank you once again for the affirmative review our work.

Round 2

Reviewer 1 Report

Dear Authors

I notice you have addressed not only my concerns but also those pointed out by other reviewers. As a result, your manuscript seems to be in better shape.